# BiCert: A Blinear Mixed Integer Programming Formulation for Precise Certified Bounds Against Data Poisoning Attacks

## Abstract

Data poisoning attacks pose one of the biggest threats to modern AI systems, necessitating robust defenses. While extensive efforts have been made to develop empirical defenses, attackers continue to evolve, creating sophisticated methods to circumvent these measures. To address this, we must move beyond empirical defenses and establish provable certification methods that guarantee robustness. This paper introduces a novel certification approach using Bilinear Mixed Integer Programming (BMIP) to compute sound, deterministic bounds that provide such provable robustness. Using BMIP, we compute the reachable set of parameters that could result from training with potentially manipulated data. A key insight to make this computation feasible is relaxing the reachable parameter set to a convex set between training iterations. At test time, this parameter set allows us to predict all possible outcomes, guaranteeing robustness. Our BMIP approach is more precise than previous methods, which rely solely on interval and polyhedral bounds. Crucially, it overcomes the fundamental limitation of prior approaches where parameter bounds could only grow, often uncontrollably. We show that these tighter bounds eliminate a key source of divergence issues, resulting in more stable training and higher certified accuracy.

## 1 Introduction

Date poisoning attacks are one of the most significant threats to the integrity of machine learning models today. Attackers can potentially exert far-reaching influence by poisoning ubiquitous foundation models that are widely used, or targeting systems in critical applications, such as finance, healthcare, or autonomous decision-making. These attacks aim to inject malicious data into the training process, leading the model to make incorrect or harmful predictions during deployment. The recent trend of training foundation models on massive datasets scraped from all available sources has exacerbated this threat, as data curation on this scale is nearly impossible (Wan et al., 2023; Carlini et al., 2024). Government agencies across the US and Europe have recognized poisoning as one of the fundamental threats to AI systems and highlight the need for robust defenses in their respective reports (ENISA, 2020; Vassilev et al., 2024) and legislation (European Union, 2024).

Identifying this threat, many empirical defenses have been developed to mitigate the risks (Cinà et al., 2023). These approaches typically use a combination of data filtering, robust training, and model inspection to detect or prevent the influence of poisoning attacks. However, their empirical nature means they can detect the presence of specific attacks but cannot guarantee their absence. As a result, more sophisticated attacks have been developed to circumvent these defenses (Suciu et al., 2018). A recent line of work explores rigorous, provable guarantees that bound the influence that poisoning attacks can have on a model's predictions (Lorenz et al., 2024; Sosnin et al., 2024). These methods are based on test-time certifiers (Gowal et al., 2018; Zhang et al., 2018) and use interval and polyhedral constraints to compute upper and lower real-valued bounds for each parameter during training. These parameter intervals bound the influence that data perturbations can have on the trained model. These works are crucial, as they are the first to provide sound, deterministic bounds on the model parameters and, consequently, an adversary's influence on the training.

However, while these types of bounds have been shown to be a good trade-off between precision and computational efficiency for test-time certification (Li et al., 2023), the significant over-approximations of polyhedral constraints limit the method's scalability for training. Lorenz et al. (2024) have analyzed this limitation and show that interval bounds can cause the training to diverge.

This work addresses the imprecisions of prior methods by proposing a more precise certification method BiCert. We formulate certified training as an optimization problem with bilinear mixed integer constraints, which can be solved without over-approximations. Since solving an optimization problem across the entire computational graph of training is computationally infeasible even for small models, we strategically cut the computational graph after each parameter update, solving the optimization problem in parameter space. We demonstrate that the bounds achieved by this method remain sound while maintaining computational feasibility, and solve the fundamental limitations of prior methods. An experimental evaluation validates our method, confirming our theoretical analysis and comparing it to prior work. The results show state-of-the-art performance, significantly outperforming the baselines for larger perturbation radii and demonstrating improved training stability.

## 2    RELATED WORK

There are two major lines of work on certified defenses against training-time attacks: *ensemble-based methods* and *bound-based* methods.

**Ensemble-Based Methods.**    Ensemble-based methods are typically based on either bagging or randomized smoothing. Wang et al. (2020), Rosenfeld et al. (2020), and Weber et al. (2023) extend Randomized Smoothing (Cohen et al., 2019) to training-time attacks. While Wang et al. (2020) and Weber et al. (2023) compute probabilistic guarantees against $\ell_0$ and $\ell_2$-norm adversaries respectively, Cohen et al. (2019) provide these guarantees against label flipping attacks.

A similar line of work provides probabilistic guarantees against training-time attacks using bagging. Jia et al. (2021) find that bagging's data sub-sampling shows intrinsic robustness to poisoning. Wang et al. (2022) enhance the robustness guarantees with advanced sampling strategies, while Levine & Feizi (2021) introduce a deterministic bagging method. Zhang et al. (2022) adapt this approach for backdoor attacks with triggers. Recent studies also explore different threat models, including temporal aspects Wang & Feizi (2023) and dynamic attacks Bose et al. (2024).

**Bound-Based Methods.**    In contrast to these ensemble-based sampling methods, Lorenz et al. (2024) and Sosnin et al. (2024) propose to compute sound, deterministic bounds of the model's parameters during training. Both methods share the same underlying principle. They define a polytope of allowed perturbations in input space and propagate it through the forward and backward passes during training. By over-approximating the reachable set with polytopes along the way, they compute sound, worst-case bounds for the model's gradients. Using these bounds, the model parameters can be updated with sound upper and lower bounds, guaranteeing that all possible parameters resulting from the data perturbations lie within these bounds. Lorenz et al. (2024) use intervals to represent these polytopes and extend the approach to also include test-time perturbations. Sosnin et al. (2024) use a combination of interval and linear bounds and additionally limit the number of data points that can be perturbed. Both approaches' biggest limitations are the significant over-approximations caused by using polytopes to represent the reachable sets, which can lead to loose bounds and divergence issues.

## 3    CERTIFIED TRAINING USING BILINEAR MIXED INTEGER PROGRAMMING

The goal of our method is to bound the error that bounded perturbations to the training data can introduce on the final machine learning model. However, precisely computing these bounds is infeasible even for small model sizes, as it has been shown that this is a $\Sigma_2^P$-hard problem (Marro & Lombardi, 2023). The challenge therefore lies in finding a feasible over-approximation, which (i) ensures sound bounds, (ii) makes as few over-approximations as possible, and (iii) remains computationally feasible.

Previous works (Lorenz et al., 2024; Sosnin et al., 2024) compute sound bounds using Interval Bound Propagation (IBP). However, their scalability is limited due to significant over-

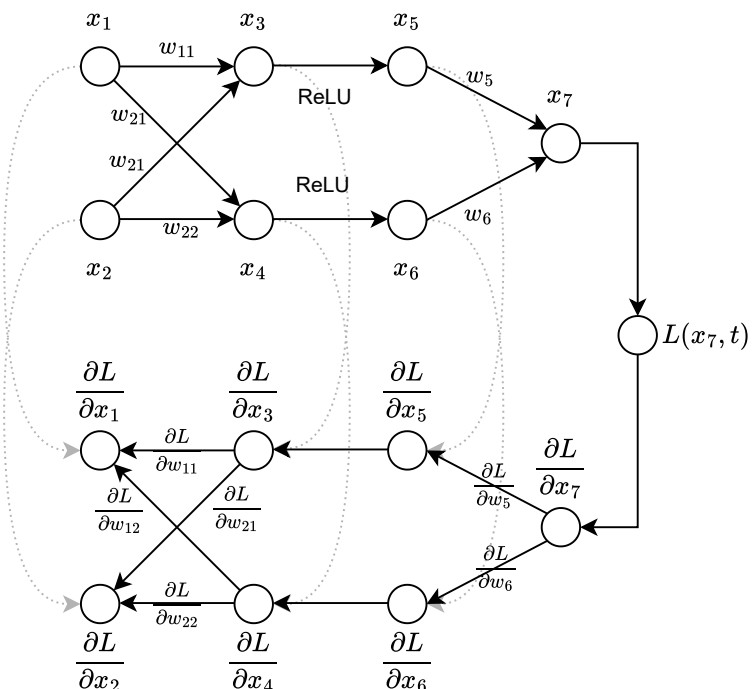

Figure 1: Illustration of a single forward and backward pass for a simplified model.

approximations of interval bounds. While over-approximations are undesirable in any certification task, they are especially costly during training. Lorenz et al. (2024) show that over-approximations can cause the training to diverge, making it prudent to minimize them.

There are two leading causes of divergence when training with interval bounds: (i) IBP does not preserve the relationship between input and output variables and therefore over-approximates even simple operations such as addition and subtraction. (ii) The size of bounds for parameters can only grow due to the interval subtraction in the parameter update step (section 4). This motivates the main design decisions of our method using Bilinear Mixed Integer Programming. BMIP allows us to compute exact bounds for one training iteration, including forward pass, loss, backpropagation, and parameter updates. In an ideal world with unlimited compute, we would propagate these exact bounds throughout the entire training and inference process. However, this is computationally infeasible. We therefore relax the bounds for the model parameters to a convex set after each training iteration. This cuts the verification graph to a single training iteration while avoiding the divergence issues of previous methods, as we will show in section 4.

## 3.1 ILLUSTRATIVE EXAMPLE

Before we formally define BiCert, we illustrate it on a simplified example (fig. 1). This illustration employs a model with only two inputs, $x_1$ and $x_2$. The first operation is a fully connected layer (without bias) with weights $w_{11}$, $w_{12}$, $w_{21}$, and $w_{22}$. This linear layer is followed by a ReLU non-linearity and a second fully connected layer with weights $w_5$ and $w_6$. We use $(x_1 = 1, x_2 = 1)$ as an example input with target label $t = 1$. We set the perturbations $\epsilon = 1$, which leads to upper and lower bounds of $[0, 2]$ for both inputs. We color all equations that belong to the optimization problem in teal. The full optimization problem with all constraints is in appendix A.

**Forward Pass.** The first step is to encode the forward pass through the model as constraints of the optimization problem. We start with the input by defining $x_1$ and $x_2$ as variables of the optimization problem. By the definition of the threat model, their constraints are $0 \le x_1 \le 2$ and $0 \le x_2 \le 2$.

For the fully connected layer, we encode the variables $x_3$ and $x_4$ with respect to $x_1$ and $x_2$. We need the parameters $w_{ij}$ as variables to do so, which are set to their initial values $w_{11} = 1$, $w_{12} = 1$, $w_{21} = 1$, and $w_{22} = -1$. One might be tempted to substitute these variables with their values to simplify the optimization problem. However, this would cause information loss, which would decrease the precision of the final solution. This effect will amplify in later rounds where $w_{ij}$ are no longer single values but intervals with upper and lower bounds. We add the constraints $x_3 = w_{11}x_1 + w_{21}x_2$ and $x_4 = w_{12}x_1 + w_{22}x_2$ to the optimization problem. Since we multiply two variables with each other, the problem becomes bilinear, which is one of the reasons we require a bilinear optimization problem.

The ReLU non-linearity can be directly encoded as a piecewise linear constraint: $x_5 = \max(0, x_3)$, or, equivalently, $x_5 = x_3$ if $x_3 > 0$, and $x_5 = 0$ otherwise. $x_6$ is defined accordingly. ReLUs are the main reason we require Mixed Integer Programming, as they allow us to encode piecewise linear constraints using binary decision variables. $x_7$ is a linear combination of the two outputs, adding the constraint $x_7 = w_5x_5 + w_6x_6$ with $w_5 = -1$ and $w_6 = 1$. At this stage, we can already see that BiCert's bounds are more precise. Computing the bounds for $x_7$ using FullCert's IBP (Lorenz et al., 2024), we get $-4 \leq x_7 \leq 2$, while solving the optimization problem gives $-4 \leq x_7 \leq 0$ with a tighter upper bound.

**Loss.** After the forward pass through the model, we encode the loss function as constraints to the optimization problem. We use the hinge loss $L(x_7, t) = \max(0, 1 - tx_7) = \max(0, 1 - x_7)$ in this example because it is a piecewise linear function and therefore can be exactly encoded, analogous to ReLUs. General losses can be supported by bounding the function with piecewise linear bounds.

**Backward Pass.** For the backward pass, we need to compute the loss gradient for each parameter using the chain rule. It starts with the last layer, which is $\frac{\partial L}{\partial x_7} = -1$ if $x_7 \leq 1, 0$ otherwise. This is also a piecewise linear function and can be encoded as a constraint to the optimization problem.

The gradients for the linear layer $x_7 = w_5x_5 + w_6x_6$ can be determined using the chain rule: $\frac{\partial L}{\partial w_5} = x_5\frac{\partial L}{\partial x_7}$ and $\frac{\partial L}{\partial x_5} = w_5\frac{\partial L}{\partial x_7}$, with corresponding expressions for $x_6$ and $w_6$. Given that the outer gradient $\frac{\partial L}{\partial x_7}$, as well as $x_5$ and $w_5$, are variables, this backward propagation leads to bilinear constraints. The derivatives of ReLU are piecewise linear, resulting in $\frac{\partial L}{\partial x_3} = 0$ if $x_3 \leq 0$ and $\frac{\partial L}{\partial x_3} = \frac{\partial L}{\partial x_5}$ otherwise. The derivatives for the parameters $w_{11}$, $w_{12}$, $w_{21}$, and $w_{22}$ function analogously to $w_5$ and $w_6$.

**Parameter Update.** The last step is the parameter update. We also encode the new parameters as a constraint: $w_i' = w_i - \lambda\frac{\partial L}{\partial w_i}$. Theoretically, we could directly continue with the next forward pass, using the new parameters $w_i'$, resulting in an optimization problem that precisely encodes the entire training. However, this is computationally infeasible in practice. We therefore relax the constraints after each parameter update by solving the optimization problem for each parameter: $\underline{w}_i' = \min w_i'$, and $\overline{w}_i' = \max w_i'$, subject to the constraints which encode the forward and backward passes from above. $\underline{w}_i'$ and $\overline{w}_i'$ are real-valued constraints that guarantee $\underline{w}_i' \leq w_i' \leq \overline{w}_i'$. This leads to valid bounds for all parameters in consecutive iterations, as mentioned above.

## 3.2 FORMAL DEFINITION

Before we introduce the details of our method, we must first define the training-time certification problem and establish the soundness of the general approach. To certify the robustness of the model's training, we have to define a precondition for which the certificate should hold. For training-time certification, this is the formalization of the adversary's capabilities to perturb the training data. We define all valid dataset permutations as a family of datasets, similar to prior work (Eq. (2) in Lorenz et al. (2024) and Section 3.1 in Sosnin et al. (2024)):

**Definition 1 (Family of Datasets)** *A family of datasets $\mathcal{D}_p^\epsilon$ is the set of all datasets that can be produced by perturbing data points $x$ from the original dataset $D$ within an $\ell_p$ ball:*

$$\mathcal{D}_p^\epsilon := \left\{ D' \mid \forall (x', y') \in D' \exists (x, y) \in D \land \|x' - x\|_p \leq \epsilon \land y' = y \right\}. \tag{1}$$

For continuous domains of $x$ (e.g., $\mathbb{R}^d$), the cardinality of $\mathcal{D}_p^\epsilon$ is generally infinite.

The final certificate should guarantee that at test time, the model's predictions are independent of the training data perturbation. Given a gradient-based training algorithm $A$ (e.g., SGD), dataset $D$, and model $f$ with parameters $\theta$, we can formalize training as $\theta = A(D, \theta_0)$, where $\theta_0$ is the parameter initialization. This allows us to define the certificate as follows:

**Definition 2 (Certificate)** *For test input $x$ and its label $y$, the certificate guarantees that*

$$f_{\theta'}(x) = y \ \forall D' \in \mathcal{D}_p^\epsilon \ \wedge \ \theta' = A(D', \theta_0). \tag{2}$$

The equation holds if we can guarantee that for all perturbed datasets in $\mathcal{D}_p^\epsilon$ the model predicts the same label.

Solving this problem is, while theoretically possible, typically infeasible in practice. The computational graph of training a model is deep, especially for larger models and datasets. We therefore divide the problem into two steps: (1) find sound bounds on the model's parameters after training, and (2) based on these parameter bounds, check whether the prediction is correct for all parameters within those bounds. This division has an additional bonus: because the training, which is the most costly part of the computation, is independent of the test sample $x$, we only have to compute the parameter bounds once and can then re-use them for multiple test samples.

**Definition 3 (Parameter Bounds)** *To formalize this, we define bounds in parameter space that contain all parameter configurations that could result from training on a dataset in $\mathcal{D}_p^\epsilon$:*

$$\underline{\theta} \leq A(D', \theta_0) \leq \overline{\theta} \ \forall D' \in \mathcal{D}_p^\epsilon. \tag{3}$$

Instead of solving eq. (2), we can then solve the simplified version:

**Proposition 1** *For valid parameter bounds $\left[\underline{\theta}, \overline{\theta}\right]$ according to definition 3, the following equation implies that definition 2 holds:*

$$f_{\theta'}(x) = y \ \forall \theta' \in \left[\underline{\theta}, \overline{\theta}\right]. \tag{4}$$

Proof: it holds that $\forall D' \in \mathcal{D}_p^\epsilon \wedge \theta' = A(D', \theta_0) \implies \theta' \in \left[\underline{\theta}, \overline{\theta}\right]$ according to definition 3. Substituting $\theta'$, eq. (4), therefore implies that eq. (2) holds. Note that the inverse is not necessarily true, as definition 3 does not require the bounds to be tight.

Splitting the test-time inference makes the trained model reusable and the test-time inference computationally feasible. However, the computational graph for model training is still too deep, as it spans multiple iterations of forward and backward passes. To solve this, we further split the problem after each parameter update by computing parameter bounds as before. We refine eq. (3) and consider bounds $\left[\underline{\theta}_i, \overline{\theta}_i\right]$ after iteration $i$ of training algorithm $A$:

$$\underline{\theta}_i \leq A_i\left(D', \theta'\right) \leq \overline{\theta}_i \ \forall D' \in \mathcal{D}_p^\epsilon, \theta' \in \left[\underline{\theta}_{i-1}, \overline{\theta}_{i-1}\right], \tag{5}$$

with $\underline{\theta}_0 = \theta_0 = \overline{\theta}_0$ and $A_i$ being the $i$-th iteration of training algorithm $A$. This relaxation is sound for all $i$ by construction, and proposition 1 holds for the bounds after each iteration. In the end, this results in $n + 1$ problems to be solved, where $n$ is the number of iterations of the training algorithm.

### 3.3 BiCert: Bounds via Bilinear Mixed Integer Programming

The key innovation of our method BiCert is the approach to solving eq. (5). Using Bilinear Mixed Integer Programming, we can compute an exact solution for each iteration, avoiding over-approximations. For each training iteration, we build an optimization problem over each model parameter, with the new, updated value as the optimization target. $\mathcal{D}_p^\epsilon$, the current model parameters, the transformation functions of each layer, the loss, the gradients of the backward pass, and the parameter update are encoded as constraints. This leads to $2 * m$ optimization problems for a model

with $m$ parameters of the form

$$
\begin{aligned}
\text{min/max} \quad & \theta^j_{i+1}, \quad j = 1, \ldots, m \\
\text{subject to} \quad & \text{Input Constraints} \\
& \text{Parameter Constraints} \\
& \text{Layer Constraints} \\
& \text{Loss Constraints} \\
& \text{Gradient Constraints} \\
& \text{Parameter Update Constraints.}
\end{aligned}
\tag{6}
$$

The objectives are the parameters, which we maximize and minimize independently to compute their upper and lower bounds. The constraints are the same for all parameters and only have to be constructed once. We present these constraints for fully-connected models with ReLU activation below.

**Input Constraints.** The first set of constraints encodes the allowed perturbations, in this case, the $\ell_\infty$ norm with radius $\epsilon$, where $o^{(0)}_k$ are the auxiliary variables encoding the $n$ input features:

$$
o^{(0)}_k \leq x_k + \epsilon, \quad k = 1, \ldots, n \qquad o^{(0)}_k \geq x_k - \epsilon, \quad k = 1, \ldots, n.
\tag{7}
$$

**Parameter Constraints.** Parameters have bounds starting from the second iteration, as discussed above. We encode them as:

$$
\theta^j_i \leq \overline{\theta}^j_i, \quad j = 1, \ldots, m \qquad \theta^j_i \geq \underline{\theta}^j_i, \quad j = 1, \ldots, m.
\tag{8}
$$

**Linear Layers.** Linear layer constraints are straight-forward, as they are linear combinations of the layer's inputs $o^{(l-1)}_v$, and the layer's weights $w^{(l)}_{uv} \in \theta_{i-1}$ and biases $b^{(l)}_u \in \theta_{i-1}$:

$$
o^{(l)}_u = \sum_v w^{(l)}_{uv} o^{(l-1)}_v + b^{(l)}_u, \quad u = 1, \ldots, \left| o^{(l)} \right|.
\tag{9}
$$

This results in bilinear constraints, as the layer's parameters are multiplied by the inputs, both of which are variables of the optimization problem.

**ReLU Layers.** ReLUs are encoded as piecewise-linear constraints, e.g., via Big-M or SOS:

$$
o^{(l)}_i = \begin{cases} 0 & \text{if } o^{(l-1)}_u \leq 0 \\ o^{(l-1)}_u & \text{otherwise} \end{cases}, \quad u = 1, \ldots, \left| o^{(l)} \right|.
\tag{10}
$$

**Loss Function.** We use the margin loss because it is piecewise linear, and we can therefore encode it exactly. For other loss functions, we can use (piecewise) linear relaxations. With the last-layer output $o^{(L)}$, the ground-truth label $y$, and auxiliary variable $J$, we define the constraint as eq. (11):

$$
J = \max\left(0, 1 - yo^{(L)}\right) \tag{11} \qquad \frac{\partial J}{\partial o^{(L)}} = \begin{cases} -y & \text{if } yo^{(L)} \leq 1 \\ 0 & \text{otherwise} \end{cases} \tag{12}
$$

**Loss Gradients.** The derivative of the margin loss is also piecewise linear (eq. (12)).

**ReLU Gradients.** The local gradient of the ReLU function is also piecewise linear (eq. (13)). Multiplication with the upstream gradient results in a piecewise bilinear constraint (eq. (14)).

$$
\frac{\partial x^{(l)}_i}{\partial x^{(l-1)}_i} = \begin{cases} 0 & \text{if } x^{(l-1)}_i \leq 0 \\ 1 & \text{otherwise} \end{cases} \tag{13} \qquad \frac{\partial L}{\partial x^{(l-1)}_i} = \frac{\partial L}{\partial x^{(l)}_i} \frac{\partial x^{(l)}_i}{\partial x^{(l-1)}_i} \tag{14}
$$

**Linear Gradients.** All partial derivatives for linear layers are bilinear:

$$
\frac{\partial J}{\partial o^{(l-1)}_u} = \sum_v w_{uv} \frac{\partial L}{\partial o^{(l)}_v} \quad \text{(15a)} \qquad \frac{\partial J}{\partial w^{(l)}_{uv}} = o^{(l-1)}_u \frac{\partial J}{\partial o^{(l)}_v} \quad \text{(15b)} \qquad \frac{\partial J}{\partial b^{(l)}_u} = o_u \frac{\partial J}{\partial o^{(l)}_u} \quad \text{(15c)}
$$

---

**Algorithm 1:** BMIP Training

---

**Input:** Dataset $D$, Initial parameters $\theta_0 \in \mathbb{R}^m$, Perturbation size $\epsilon \in \mathbb{R}$, Num iterations $n \in \mathbb{N}$

**Output:** Parameter bounds $\underline{\theta}_n \in \mathbb{R}^m$, $\overline{\theta}_n \in \mathbb{R}^m$

1 Initialize $\underline{\theta}_0, \overline{\theta}_0 \leftarrow \theta_0$;
2 **for** $i = 1$ *to* $n$ **do**
3 $\quad$ bmip $\leftarrow$ *initialize_optimization_problem*();
4 $\quad$ bmip.*add_parameter_constraints*($\underline{\theta}_{i-1}, \overline{\theta}_{i-1}$);
5 $\quad$ **for** $x \in D$ **do**
6 $\quad\quad$ bmip.*add_input_constraints*($x, \epsilon$);
7 $\quad\quad$ **for** *each layer* $l = 1$ *to* $L$ **do**
8 $\quad\quad\quad$ bmip.*add_layer_constraints*($l$);
9 $\quad\quad$ bmip.*add_loss_constraints*();
10 $\quad\quad$ bmip.*add_loss_gradient_constraints*();
11 $\quad\quad$ **for** *each layer* $l = L$ *to* $1$ **do**
12 $\quad\quad\quad$ bmip.*add_layer_gradient_constraints*($l$);
13 $\quad$ bmip.*add_parameter_update_constraints*();
14 $\quad$ $\underline{\theta}_i =$ bmip.*minimize*($\theta_i$);
15 $\quad$ $\overline{\theta}_i =$ bmip.*maximize*($\theta_i$);
16 **return** $\underline{\theta}_n$, $\overline{\theta}_n$;

---

**Parameter Updates.** The last set of constraints is the parameter updates. It is essential to include this step before relaxation because the old parameters are contained in both subtraction operands. Solving this precisely is a key advantage compared to prior work (section 4).

$$\theta_{i+1}^j = \theta_i^j - \lambda \frac{\partial J}{\partial \theta^j}, \quad j = 1, \ldots, m. \tag{16}$$

### 3.4 ALGORITHMIC IMPLEMENTATION

We implement this optimization procedure according to algorithm 1, effectively unrolling the training procedure. For each iteration of the training algorithm (line 2), we initialize an optimization problem (line 3) and add the current parameter bounds as constraints (line 4). For each data point $x$, we add the input constraints (line 6), layer constraints (line 8), loss constraints (line 9), derivative constraints (lines 10-12), and parameter update constraints (line 13). We then solve the optimization problem twice for each parameter, once for the upper and once for the lower bound (lines 14-15). The algorithm returns the final parameter bounds.

Once the model is trained, we can use the final parameter bounds for prediction (algorithm 2). The principle is the same as encoding only the forward pass from training (lines 1-5). For classification, we can then compare the logit and check whether one is always greater than all others (lines 6-12). If so, we return the corresponding class (line 12). Otherwise, we cannot guarantee a prediction, and the algorithm has to *abstain* (line 13). The post-condition (lines 6-13) can be adjusted, e.g., for regression tasks.

## 4 THEORETICAL COMPARISON TO PRIOR WORK

As discussed before, the main limitation of prior work is the significant over-approximations introduced by relaxing bounds to interval/polyhedral constraints after each layer. While these techniques have shown an acceptable trade-off for inference-time certification (Gowal et al., 2018; Boopathy et al., 2019; Singh et al., 2019), they require robust training of the underlying models to account for and minimize these over-approximations (Mao et al., 2024).

For training-time certification, these trade-offs are fundamentally different. For one, the over-approximations cannot be compensated for, as using the bounds as loss targets would invalidate the guarantees. Furthermore, the computational graph is significantly deeper, so the over-

---

**Algorithm 2:** BMIP Predict

---

**Input:** Test data $x$, Parameter bounds $\underline{\theta}, \overline{\theta} \in \mathbb{R}^m$
**Output:** Certified prediction $y$, or *abstain*

1  bmip $\leftarrow$ *initialize_optimization_problem*();
2  bmip.*add_parameter_constraints*($\underline{\theta}, \overline{\theta}$);
3  bmip.*add_input_variables*($x$);
4  **for** *each layer $l = 1$ to $L$* **do**
5       bmip.*add_layer_constraints*($l$);

6  **for** *each logit $o_u^{(L)}$* **do**
7       $c \leftarrow$ True;
8       **for** *each logit $o_v^{(L)} \neq o_u^{(L)}$* **do**
9           $c_v =$ bmip.*minimize*($o_u^{(L)} - o_v^{(L)}$);
10          $c \leftarrow c \wedge (c_v \geq 0)$;
11      **if** $c$ **then**
12          **return** $u$;

13 **return** *abstain*;

---

approximations accumulate more. Lorenz et al. (2024) show that these accumulating over-approximations pose fundamental barriers for the training to converge, often leading to exploding bound sizes, even at a (local) minimum.

To analyze this behavior, Lorenz et al. (2024) use the Lyapunov sequence $h_i = (\theta_i - \theta^*)^2$ that measures the distance of the current parameter vector $\theta_i$ to the optimum $\theta^*$, as originally proposed by Bottou (1998). Under some assumptions (Bottou, 1998), we can show that SGD converges if $h_i$ converges. To do so, we expand the sequence to eq. (17) (eq. 15 in (Lorenz et al., 2024)) using the parameter update step.

$$h_{i+1} - h_i = \underbrace{-2\lambda_i(\theta_i - \theta^*)\nabla_\theta J(\theta_i)}_{\text{distance to optimum}} + \underbrace{\lambda_i^2 (\nabla_\theta J(\theta_i))^2}_{\text{discrete dynamics}}. \tag{17}$$

The second term relating to the discrete nature of the algorithm is bounded by an assumption on a monotone decrease in the learning rate (mostly strict convexity and bounded gradients, see Bottou (1998)). It remains to show that the first term, which contains the distance to the optimum multiplied by the gradient and the negative learning rate, is negative. Bottou (1998) show that this is the case for regular SGD, where $\theta$ is a vector of real numbers. Lorenz et al. (2024) show that for intervals $\Theta$, this holds if $\Theta_i \cap \Theta^* = \emptyset$, that is, the algorithm diverges if the optimum intersects with the current parameters.

We show that this limitation does not apply to exact certifiers. By definition, the solution does not contain any over-approximations in the update step. Therefore, it holds that

$$-2\lambda_i(\Theta_i - \Theta^*)\nabla_\theta J(\Theta_i) \geq 0 \iff \exists \theta_i \in \Theta_i, \theta^* \in \Theta^* - 2\lambda_i(\theta_i - \theta^*)\nabla_\theta J(\theta_i) \geq 0 \tag{18}$$

Since the right-hand side is false according to Bottou (1998), it implies that the term is negative, and therefore our approach does not suffer from the same limitation.

The second limitation Lorenz et al. (2024) show is the fact that the size of the parameter intervals can only increase (their eq. 16). This is related to the parameter update step $\theta_{i+1} = \theta_i - \lambda \nabla_\theta J(\theta_i)$. If both operands of the subtraction are intervals, the size of the resulting interval is the sum of both sizes, i.e., $|\theta_{i+1}| = |\theta_i| + |\lambda \nabla_\theta J(\theta_i)|$. In contrast, our exact solution allows parameter intervals to shrink as we preserve the relationship between the two operands (eq. (16)).

Solving these two fundamental limitations, combined with generally significantly more precise bounds, solves many of the training instabilities of prior work. In particular, we did not encounter any diverging bounds where training fails due to bounds approaching $\pm\infty$. This increased precision comes at the cost of increased training time, as solving the BMIP is more expensive than loose interval bounds.

| $\epsilon$ | 0.0001 | 0.001 | 0.01 |
|---|---|---|---|
| FullCert | 83.9% $\pm$ 3.60 | 82.2% $\pm$ 4.40 | 71.5% $\pm$ 11.20 |
| Sosnin et al. | **85.6%** $\pm$ **0.04** | **84.0%** $\pm$ **0.04** | 44.9% $\pm$ 0.22 |
| BiCert (Ours) | 83.3% $\pm$ **0.05** | 82.0% $\pm$ **0.05** | **81.4%** $\pm$ **0.06** |

Table 1: Comparison of BiCert to FullCert (Lorenz et al., 2024) and Sosnin et al. (2024) across different $\epsilon$ values. The numbers represent the mean and standard deviation of certified accuracy across randomly chosen seeds.

## 5 EXPERIMENTS

We perform several experiments to evaluate our method and to validate our theoretical findings. To this end, we compare BiCert to prior work and analyze its runtime behavior.

### 5.1 IMPLEMENTATION DETAILS

We use Gurobi (Gurobi Optimization, LLC, 2024) as a solver for this work. It offers several benefits, including support for piecewise linear and bilinear constraints and a Python interface for integrating deep learning frameworks. The library also supports reusing the same optimization model for multiple optimizations, including warm starts that allow for more efficient solutions. To represent the parameter bounds, we use the BoundFlow library by Lorenz et al. (2024). Our implementation also supports mini-batch processing, similarly to typical deep learning frameworks. Additional implementation and training details, including model architectures, hyperparameters, and hardware configuration, are listed in appendix B.

### 5.2 CERTIFIED ACCURACY ACROSS PERTURBATION SIZES

To validate the theoretical analysis of our method, we evaluate BiCert experimentally and compare it to FullCert (Lorenz et al., 2024) and Sosnin et al. (2024). We evaluate it on the Two-Moons dataset for classification, a widely used dataset with two classes of points configured in interleaving half circles. Table 1 presents the certified accuracy, i.e., the percentage of data points from a held-out test set where algorithm 2 returns the ground-truth class. All values are mean and standard deviation across 5-10 runs with different random seeds.

For small perturbation radii $\epsilon$, our method performs similarly to the baselines. With increasing radius, the advantages of tighter bounds become apparent, where our method significantly outperforms the baselines. This trend makes sense, as the over-approximations become more influential with larger perturbations. The second advantage of our method becomes apparent when looking at the standard deviation, especially for FullCert and for larger perturbations. The small standard deviation shows a much more stable training behavior compared to the baseline, which aligns with our analysis in section 4.

### 5.3 RUNTIME AND COMPLEXITY ANALYSIS

Of course, there is no free lunch when it comes to optimization. In our case, the higher precision and, therefore, certified accuracy, comes at the cost of an increased computational complexity. We investigate this impact by measuring the time it takes to train our models. Table 2 shows the average runtime for each perturbation size for the first 10 epochs. The general trend is an increased complexity for later epochs and an increased complexity for larger perturbation radii.

These results are consistent with our expectations. Larger perturbations increase the problem complexity, as they give the adversary a higher degree of freedom and increase the set of possible parameters. More iterations also increase the influence of those perturbations, as the data is processed multiple times, in addition to the effects of accumulating over-approximations as the depth of the computational graph grows.

| Epoch | $\epsilon = 0.0001$ | $\epsilon = 0.001$ | $\epsilon = 0.01$ |
|---|---|---|---|
| 1 | 8 | 9 | 9 |
| 2 | 38 | 65 | 112 |
| 3 | 15 | 66 | 141 |
| 4 | 19 | 65 | 199 |
| 5 | 14 | 65 | 321 |
| 6 | 21 | 73 | 719 |
| 7 | 21 | 87 | 1023 |
| 8 | 23 | 103 | 2626 |
| 9 | 28 | 162 | 6711 |
| 10 | 29 | 299 | 12502 |

Table 2: Average runtime per epoch for different $\epsilon$ in seconds. The runtime increases for larger perturbations and later epochs.

## 6 DISCUSSION

BiCert addresses the fundamental limitations of prior certified training approaches, specifically the over-reliance on convex over-approximations that lead to unstable training and diverging parameter bounds. By introducing Bilinear Mixed Integer Programming (BMIP) for certified training, we provide tighter, more precise bounds at each training iteration. Our method ensures that the parameter bounds can shrink, effectively mitigating the divergence issues faced by previous methods.

The primary benefit of this approach lies in its improved precision, which leads to significantly higher certified accuracy, especially for larger perturbations. As demonstrated in our experiments, BiCert consistently outperforms existing methods in terms of the certified accuracy of the model. Additionally, the reduced variance in our results highlights the robustness and reliability of our approach.

This increased precision comes at the cost of a higher computational complexity. Solving piecewise bilinear optimization problems for each parameter update is more computationally expensive than the interval-based methods used in prior work. This trade-off between computational cost and certification precision is crucial when deploying certified defenses in practical settings. While the added cost may be prohibitive for large-scale models, our method provides a valuable framework for improving the precision and reliability of certification in smaller models or scenarios where robustness is critical.

Looking forward, there are opportunities to explore hybrid approaches that combine the precision of BMIP with the computational efficiency of interval-based methods, potentially achieving a more scalable solution. Intelligently choosing the best constraints to relax while preserving precision where required could combine the best of both worlds, yielding precise, scalable bounds.

Overall, this work provides a significant advancement in certified defenses, demonstrating that more precise certification methods are feasible and necessary for ensuring the robustness of machine learning models against poisoning attacks. Despite the computational cost, our approach offers a substantial improvement in certified accuracy and stability, laying the groundwork for future research in precise, scalable certification techniques.

## 7 CONCLUSION

This work addresses fundamental limitations in state-of-the-art certifiers that compute bounds for training-time attacks on neural networks. In particular, their over-reliance on convex over-approximations limits their applicability to larger perturbation sizes, makes training unstable, and can cause models to diverge even for ideal, convex settings. Using Bilinear Mixed Integer Programming (BMIP), we can compute exact bounds for each training iteration. This approach significantly improves training stability and certified accuracy of the resulting models, outperforming the state-of-the-art. We believe that this work lays the foundation for effective certified defenses against training-time attacks.

## REPRODUCIBILITY STATEMENT

Reproducibility is a cornerstone of scientific progress, and we have taken careful steps to ensure that our results can be independently verified. All algorithms used in this work, including the BMIP training procedure (algorithm 1) and the prediction algorithm (algorithm 2), are described in detail. Additionally, the illustrative example (section 3.1) offers a step-by-step guide to understanding our method. The bounds for all layers, the loss, their gradients, and the parameter update are derived and explained in section 3.3. Our implementation and the experimental setup are explained in section 5.1 and appendix B, where we describe the solver setup, datasets, model architectures, and hyperparameters. We are committed to publishing the code for our experiments under an open-source license when the paper is published. We refrain from including it with the submission due to ICLR's policy of making supplementary material publicly available while under review.

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

APPENDIX

## A  FULL EXAMPLE

In section 3.1 we present an example model to illustrate our method. We list the full optimization problem for the first training iteration here.

$$
\begin{aligned}
\text{min/max} \quad & w_i', \quad i \in \{11, 12, 21, 22, 5, 6\} \\
\text{subject to} \quad & 0 \leq x_1 \leq 2 \\
& 0 \leq x_2 \leq 2 \\
& 1 \leq w_{11} \leq 1 \\
& 1 \leq w_{12} \leq 1 \\
& 1 \leq w_{21} \leq 1 \\
& -1 \leq w_{22} \leq -1 \\
& -1 \leq w_5 \leq -1 \\
& 1 \leq w_6 \leq 1 \\
& x_3 = w_{11}x_1 + w_{21}x_2 \\
& x_4 = w_{12}x_1 + w_{22}x_2 \\
& x_5 = \begin{cases} x_3 & \text{if } x_3 > 0 \\ 0 & \text{if } x_3 = 0 \end{cases} \\
& x_6 = \begin{cases} x_4 & \text{if } x_4 > 0 \\ 0 & \text{if } x_4 = 0 \end{cases} \\
& x_7 = w_5 x_5 + w_6 x_6 \\
& L = \begin{cases} 1 - x_7 & \text{if } 1 - x_7 > 0 \\ 0 & \text{otherwise} \end{cases} \\
& \frac{\partial L}{\partial x_7} = \begin{cases} -1 & \text{if } 1 - x_7 > 0 \\ 0 & \text{otherwise} \end{cases} \\
& \frac{\partial L}{\partial w_6} = x_6 \frac{\partial L}{\partial x_7} \\
& \frac{\partial L}{\partial x_6} = w_6 \frac{\partial L}{\partial x_7}
\end{aligned}
$$

$$
\begin{aligned}
& \frac{\partial L}{\partial w_5} = x_5 \frac{\partial L}{\partial x_7} \\
& \frac{\partial L}{\partial x_5} = w_5 \frac{\partial L}{\partial x_7} \\
& \frac{\partial L}{\partial x_4} = \begin{cases} \frac{\partial L}{\partial x_6} & \text{if } x_4 > 0 \\ 0 & \text{if } x_4 = 0 \end{cases} \\
& \frac{\partial L}{\partial x_3} = \begin{cases} \frac{\partial L}{\partial x_5} & \text{if } x_3 > 0 \\ 0 & \text{if } x_3 = 0 \end{cases} \\
& \frac{\partial L}{\partial w_{11}} = x_1 \frac{\partial L}{\partial x_3} \\
& \frac{\partial L}{\partial w_{12}} = x_1 \frac{\partial L}{\partial x_4} \\
& \frac{\partial L}{\partial w_{21}} = x_2 \frac{\partial L}{\partial x_3} \\
& \frac{\partial L}{\partial w_{22}} = x_2 \frac{\partial L}{\partial x_4} \\
& \frac{\partial L}{\partial x_2} = w_{21} \frac{\partial L}{\partial x_3} + w_{22} \frac{\partial L}{\partial x_4} \\
& \frac{\partial L}{\partial x_1} = w_{11} \frac{\partial L}{\partial x_3} + w_{12} \frac{\partial L}{\partial x_4} \\
& w_i' = w_i - \lambda \frac{\partial L}{\partial w_i}
\end{aligned}
$$

## B  TRAINING AND IMPLEMENTATION DETAILS

We train our models using a combination of the BoundFlow library (Lorenz et al., 2024) and the Gurobi optimizer (Gurobi Optimization, LLC, 2024). Since PyTorch (Paszke et al., 2019) does not support bound-based training, we only use its basic tensor representations.

All computations are performed on a compute cluster, which mainly consists of AMD Rome 7742 CPUs with 128 cores and 2.25 GHz. Each task is allocated up to 32 cores. No GPUs are used since Gurobi does not use them for solving.

Unless indicated otherwise, we use fully connected networks with ReLU activations, two layers, and 20 neurons per layer. For binary classification problems, we use margin loss, i.e., $J = \max(0, 1 - y \cdot f(x))$, because it is piecewise linear and can therefore be encoded exactly. It produces similar results to Binary Cross-Entropy loss for regular training without perturbations.

We train models until convergence using a held-out validation set, typically after 5 to 10 epochs on Two-Moons. We use a default batch size of 100 and a constant learning rate of 0.1. We sub-sample

the training set with 100 points per iteration. All reported numbers are computed using a held-out test set that was not used during training or hyper-parameter configuration.

For the comparison to Lorenz et al. (2024), we use the numbers from their paper. Since Sosnin et al. (2024) do not report certified accuracy for Two-Moons, we train new models in an equivalent setup. As a starting point, we use the Two-Moons configurations provided in their code. We change the model architecture to match ours, i.e., reducing the number of hidden neurons to 20. We also set $n = |D|$ to adjust the threat model to be the same as ours. The solver mode is left at its preconfigured "interval+crown" for the forward pass and "interval" for the backward pass. We ran the experiments 10 times with the same random seeds to compute mean and standard deviation statistics.

