# OpenReview forum: "BiCert: A Blinear Mixed Integer Programming Formulation for Precise Certified Bounds Against Data Poisoning Attacks"
_ICLR.cc/2025/Conference — ICLR 2025 Conference Withdrawn Submission_

### Official Review · Reviewer_coXB · 2024-10-28

**Soundness:** 2
**Presentation:** 3
**Contribution:** 1
**Rating:** 3
**Confidence:** 4

**Summary:**

The authors apply interval bound propagation style certifications at training time. This follows on from several other works published in 2024, however the authors demonstrate that their approach produces results that remain stable as the perturbation size increases.

**Strengths:**

The authors demonstrate apparent improvements over reference papers, including more stable performance, in that the standard deviation across experiments remains low relative to their two tested points of comparison.

This work also handily provides a clear indicative example of the overall process, which would be of assistance to readers who are not familiar with the basics of Interval Bound Propagation (IBP) style approaches. However (as will be noted below) it fails to contextualise what the authors have done differently for those who are familiar with IBP.

**Weaknesses:**

The following weaknesses will be divided into two sections: initially I will discuss major weaknesses that need to be highlighted, before a more general commentary covering indicative examples of issues with presentation and writing.

- Section 2 really doesn't provide much context as to why the authors have taken the approach they have taken, and would be relatively inaccessible to a reader who does not know the field.
- If the motivating threat model is that an attacker could replace data online, that would then be scraped and ingested into a model to induce a poisoning attack - why would an attacker maintain an \ell_p norm bounded perturbation on their input. The threat model would suggest that all inputs would be modified, however this does not align with the actual mechanism of implementing this threat.
- Appears to only applies to ReLU style architectures - highly limited in applicability. I appreciate that the authors are following on from several 2024 works in this same space, which suggests that this is the state of the art in the area - but when randomised approaches do not need to consider such architectural limitations, then this would appear to be a significant step away from what is possible, for questionable gain.
- Perturbation radii are listed at 10^(-4), 10^(-3) and 10^(-2), seemingly without acknowledging the norm of the perturbation? These are shockingly small perturbations. Though obviously there is an impact of perturbations of size 10^(-2) because the reference certifications are affected at this size, but still, these sizes seem pointlessly small. In an image based context for example, changing a single pixel by one value (in pixel space) produces a norm of just under 0.004. That we are talking about changes below the pixel resolution suggests that perturbation inputs are not clipped to feasible inputs, making the poisoning detectable and defeatable by any naive approach.
- Computational times presented in terms of second per epoch makes it difficult to interpret how total training time would be affected. Training over 10 epochs would appear to require 6.8 hours at \epsilon = 0.01
- Computational time comparisons fail to contextualise against the reference techniques they assessed.
- Authors mention they use Gurobi, but the neural network framework and the computational architecture are only referenced in the appendices - both would be important points for understanding and assessing the value of this work as a reader.
- Two-Moons as a sole point of comparison is not within community expectations.

More general weaknesses, with a focus upon providing indicative examples of writing style issues:
- Statements like "one of the most significant threats to the integrity" are made without a) explaining what these poisoning attacks are, and b) explaining why they're a significant threat.
- More broadly the introduction tended to explain the impact of things before explaining what they are, which may make it more difficult to access by a reader who is less familiar with the field. For example data poisoning, attackers, and foundation models are all concepts that are introduced under the assumption that the reader understands them.
- in "their respective reports" - this implies that a singular report is possible from the US, and from Europe. "Both the US and Europe have respectively recognised..." would make more grammatical sense.
- S1 P2 starts off talking about detecting or preventing attacks, and then sentence two is solely focused upon detection - grammatically this doesn't flow.
- "these defences" - but at this stage the subject of the sentence is the attacks, so these is not targeted to the right subject.
- Citations for provably guarantees (see 48-49) for poisoning suggests that this is only something that has come into play in 2024, as both cites are from this year. However there are other, earlier examples, see [1] for example. Moreover there are other non interval, non polyhedral constraint approaches. One could argue that polyhedral based approaches provide deterministic bounds whereas randomised smoothing uses probabilistic (with high confidence) bounds, and this would be a reasonable point if it was explored with nuance. But by failing to acknowledge the existence of other, similar approaches the authors appear to be setting the ground for cherry picked comparisons. That said, the authors do briefly discuss these in Section 2
- L56 "limit the method's scalability for training" cite? The following sentence talks about Lorenz and training time divergence, but this is not necessarily the same issue as scalability.
- Personally I'm not a massive fan of appendix/fig/section/equation being lower case.
- Line 175 "At this stage, we can already see that BiCert's bonds are more precise." - the reader certainly can't. Your next sentence is that you test against IBP to demonstrate a tighter bound.
- Section 3.1 - the authors compare to IBP, but fail to make a strong case as to where the differences betweeen their approach and IBP stem from.
- Line 216 - if you're going to make an isolated statement about this, comments on either a) why it's important or b) when it would not be generally infinite are necessary. Otherwise it's just a single line paragraph with no meaning.
- Line 265 "is the approach to solving eq 5" - grammatically this should be "is our approach for solving". There can be multiple approaches, so a singular the approach is not correct.
- Line 411 "monotone decrease" should be "monotonically decreasing learning rate". The bracketed part of this sentence doesn't make particular sense to me as a reader.

**Questions:**

- Interval Bound Techniques can introduce some limitations on the types of architectures that are employed in models. It would appear to be that your approach only applies to ReLU style architectures - is this impression correct?
- Did you consider the possibility for any label flipping poisoning attacks?
- Is a potential bounded perturbation of all inputs (Definition 1) more realistic than entries being added/removed from the dataset, as has been explored in other certifications for poisoning attacks?
- What norm  are results like Table 1 constructed against?
- Why are your results in Table 1 run across "5-10 runs"? Why were the number of runs not consistent across the tested approaches?
- How would your approach perform at more reasonable perturbation scales?
- Why did you need to employ Gurobi as an optimizer?
- Authors note that no GPUs were used, as Gurobi doesn't employ them. Would your comparison techniques use GPUs if you made them accessible to them? IBP style solvers have been constructed on GPU's in the past - what makes your approach different?

**Details Of Ethics Concerns:**

I note no ethical concerns that warrant additional review at this time.

---

### Official Review · Reviewer_dMcf · 2024-11-03

**Soundness:** 3
**Presentation:** 2
**Contribution:** 2
**Rating:** 3
**Confidence:** 3

**Summary:**

The paper presents a certification method against data poisoning attacks named BiCert, which aims to provide bounds on model parameters under the deterministic bound setting (as opposed to high probability bounds by random smoothing) using a Bilinear Mixed Integer Programming method. While existing approaches typically rely on interval propagation or polyhedral methods, BiCert makes a convex relaxation. The authors claim the approach address the issue of uncontrollable growth of parameter bounds in prior works. They provide theoretical analysis of soundness, and experiments validation.

**Strengths:**

The need for certified robustness in machine learning is an important area, and the certified defence against training-time attacks is a critical research question. The idea of using a constrained optimization approach via bilinear MIP to addressing the problem of growing parameters bounds is interesting. The didactic illustration of the computation process of a NN is effective at helping the reader better understanding the method.

**Weaknesses:**

1. The author demonstrates the proposed method only on a basic MLP with ReLU activation and a hinge loss function, without extending it to other commonly used configurations. Specifically, there is no application shown for other activation functions (e.g., tanh, sigmoid, arctan), alternative loss functions (e.g., cross-entropy), or diverse neural structures such as convolutional, attention-based, residual, and recurrent models. This limited scope raises concerns about the broader applicability of the proposed method.
2. The experimental evaluation was conducted on a minimal binary dataset with a simple model containing only 40 neurons. To better validate the effectiveness of the method, experiments in more practical settings are needed, such as using a larger dataset like MNIST and increasing the model complexity.
3. The clean performance of the final model is not reported, and the paper lacks an ablation study examining the influence of hyperparameters. This omission makes it challenging to assess the robustness and adaptability of the method under varied parameter settings.
4. The computational overhead of the proposed method is not thoroughly evaluated, particularly as the baseline computation time without certification is not provided. The available results indicate a potentially very high overhead even in simple settings, which brings into question the practicality of this method for more complex scenarios.

**Questions:**

1. The threat model presented in Eq (1) is somewhat unclear. It appears to allow the attacker to modify each data point’s features within specific bounds, which differs from the threat models described in Lorenz et al. and Sosnin et al. Could you clarify how adjustments were made to ensure a fair comparison with these models in the experiments?
2. The method used to solve the constrained optimization problem is not clearly explained. Could you provide more details on this process?
3. How do you verify that the obtained minimum and maximum parameter values constitute certifiable bounds? In other words, how to you verify the optimization problem is correctly solved and the solutions can be used as certifiable bounds? Additional clarification on this proof would be helpful.

---

### Official Review · Reviewer_wAej · 2024-11-03

**Soundness:** 1
**Presentation:** 3
**Contribution:** 2
**Rating:** 5
**Confidence:** 4

**Summary:**

This paper introduces a novel method called BiCert, which uses bilinear mixed integer programming (BMIP) to provide exact certified parameter bounds of neural networks against (feature) poisoning attacks. The motivation of the proposal resides in the limitations of existing certification approaches, which primarily use interval and polyhedral bounds, that suffer from limited precision and often lead to instability and divergence in training due to bound over-approximation.

BiCert addresses these issues by leveraging BMIP to establish precise, deterministic bounds that ensure the robustness of the training process. Unlike previous methods that accumulate over-approximations across layers, BiCert calculates exact bounds for each training iteration. This is obtained by casting every step of a training iteration in a BMIP optimization problem and propagating the maximum and minimum value of each parameter of the network at every iteration. The provided theoretical analysis highlights that the proposed exact certification approach does not cause training to diverge and that the size of parameter intervals can also decrease over the training iterations.

The experimental methodology considers the Two-Moons dataset and a two-layer MLP with 20 neurons per layer. BiCert outperforms existing methods in terms of certified accuracy and training stability, although it comes at a higher computational cost due to the exactness of the approach.

**Strengths:**

Thanks to the authors for submitting this interesting paper that presents some strong points. First, the certification algorithm proposed in the paper is certainly novel since this is the first exact certification approach for neural networks against (feature) poisoning attacks. The paper is well written and clearly explains the building blocks at the basis of the certification algorithm, providing also an example to simplify the complex parts. Moreover, the theoretical analysis of the properties of the proposed algorithm is good and clearly sheds light on interesting properties. In particular, the certification algorithm does not cause the training of the neural network to diverge; that is a desirable property that other proposals in the literature do not grant. Finally, the experimental results about certified accuracy highlight that the proposed method provides a more precise estimation of the certified accuracy than previous proposals.

**Weaknesses:**

Even though the proposal is original and well explained, the experimental results do not greatly support the significance of the work.

While the experimental results on larger perturbations than those tested in previous work are appreciated, the considered experimental settings currently does not show the practicability of the proposed algorithm. In particular, the experiments are restricted to the Two-Moons dataset and a very small neural network (a two-layer model with only 20 neurons per layer). This setup is modest compared to the models tested by competitors. For instance, Lorenz et al. (2024) use a three-layer MLP with 40 neurons per layer and the MNIST 1/7 dataset, while Sosnin et al. (2024) includes evaluations on a convolutional neural network with MedMNIST. To demonstrate the broader applicability and significance of the proposed approach, the authors should test BiCert on larger neural networks and datasets, similar to those used in prior work. Moreover, the paper should include a comparison of the time required to certify accuracy for each method, not just the proposed method. This would allow for a more thorough assessment of the trade-off between computational cost and the precision of the analysis, that is already discussed at a high level in Section 6 without supporting experimental evidence.

**Minor points**
There are also some minor points that the authors should address in the next version of their paper, which are detailed in the Questions Section.

**Arguments for the score**
Even though the proposed certification algorithm is original, explained well, and theoretically studied, the experimental evaluation of the paper highlights that this certification algorithm can only be applied to very small neural networks. This fact undermines the significance of the proposal. I am open to increasing my score if the authors address the issues with the experimental evaluation.

**Questions:**

- Can the authors show experimental results on more realistic datasets and larger models? (See the Weaknesses Section for details).
- At the end of Section 3.2, the authors claim that $n+1$ optimization problems need to be solved if the algorithm performs $n$ iterations. Should the problems be $n$ instead?
- Why are the parameters of the neural network referred to with both $w$ and $\theta$?
- Can the authors provide more details about the motivation of `For one, the overapproximations cannot be compensated for, as using the bounds as loss targets would invalidate the guarantees.` (Section 4)?
- Can the authors provide more details about the reason why the right-hand side of proposition 18 is false (Section 4)?
- What is the number of runs performed on each experiment? The paper reports `5-10`, which is not clear. Moreover, what is the norm $p$ considered by the threat model? I guess $p=\infty$, but I am not sure about that.

**Details Of Ethics Concerns:**

.

---

### Official Review · Reviewer_F4i6 · 2024-11-04

**Soundness:** 2
**Presentation:** 2
**Contribution:** 1
**Rating:** 3
**Confidence:** 4

**Summary:**

The paper addresses the issue of poisoning attacks in machine learning, where attackers inject malicious data into a training dataset to influence a model's behavior during deployment. Traditional defenses against such attacks often rely on data filtering or robust training methods; however, these empirical approaches can be circumvented by increasingly sophisticated attacks. As a result, developing certifiable defenses that provide provable robustness guarantees has become a key priority. This paper introduces BiCert, a novel framework that strengthens certified defenses by calculating more precise bounds on model parameters through Bilinear Mixed Integer Programming. BiCert’s method eliminates training divergence and claims to achieve state-of-the-art certified accuracy, outperforming interval-propagation-based methods by ensuring greater robustness across various perturbation levels. Experimental validation on the Two-Moons demonstrates BiCert’s superior performance in certified accuracy, particularly at larger perturbation sizes.

**Strengths:**

The topic of this paper is both timely and highly relevant, with strong potential to make a meaningful contribution to the field. The paper is generally well-written, and the claimed contributions are clearly presented.

**Weaknesses:**

In my view, this paper may still be too preliminary for acceptance at a top-tier conference. Below are some points that I believe should be addressed to strengthen the work:

**1. Related Work:** The paper does not appear to fully situate itself within the existing body of literature. The primary comparisons are with Lorenz et al. (2024) and Sosnin et al. (2024); however, these works do not thoroughly cover existing research on bound propagation and mixed integer programming for adversarial robustness, which is highly relevant to this study. I recommend reviewing and comparing with additional related work, such as [this paper](https://centralesupelec.hal.science/hal-04163747/document) and references therein, to better contextualize the contributions.

**2. Problem Statement and Formalism:** The paper would benefit from a clearer, more structured problem statement. Currently, the setup is introduced incrementally (e.g., through the example in Section 3.1), without a prior definition of relevant notations and problem terms. It would improve readability to establish the notation, define key spaces and terms, and present the problem formulation explicitly before moving forward with the technical content.

**3. Practicality of the Method:** The proposed method has only been evaluated on a single, low-dimensional example (Two-Moons), which may be insufficient to convince readers of its practicality. Additional testing on benchmark datasets, such as MNIST or CIFAR, would greatly strengthen the empirical results and demonstrate the method’s broader applicability.

**4. Scalability of the Results:** While the method achieves better bounds than comparable works, it comes with higher computational costs, especially in later training epochs and with larger perturbation levels. Currently, the method’s significant computational expense, even on a simple dataset, limits its applicability. Future work might consider optimizing the computational load to make the approach more feasible for complex, real-world scenarios.

**Questions:**

While I do not have specific questions at this time, the authors could react to the above comments.

---

### Note · Authors · 2024-11-22

**Comment:**

We sincerely thank the reviewers for their thorough evaluation and constructive feedback on our work. While we do not entirely agree with all the comments and critiques raised, we recognize the overarching consensus that our paper would significantly benefit from a revision incorporating more extensive experiments exploring more realistic and practical settings and additional evaluations to contextualize our contribution better. We are encouraged that the reviewers agree that our proposed method represents an important step towards certified robustness against training-time attacks, recognizing its novelty and theoretical foundations.

In light of this, we have decided to withdraw our submission to address these points in a revised and more comprehensive version of the paper, which is beyond the scope of a simple revision. We are grateful for the reviewers’ insights, which have highlighted directions for improvement, and we look forward to resubmitting a stronger manuscript in the future.

**Withdrawal Confirmation:**

I have read and agree with the venue's withdrawal policy on behalf of myself and my co-authors.